# Feedback Modulation between Human INO80 Chromatin Remodeling Complex and miR-372 in HCT116 Cells

**DOI:** 10.3390/ijms241310685

**Published:** 2023-06-26

**Authors:** Junaid Ali Shah, Yujuan Miao, Jinmeng Chu, Wenqi Chen, Qingzhi Zhao, Chengyu Cai, Saadullah Khattak, Fei Wang, Jingji Jin

**Affiliations:** 1School of Life Sciences, Jilin University, Changchun 130012, China; junaid1316@mails.jlu.edu.cn (J.A.S.); yjmiao19@mails.jlu.edu.cn (Y.M.); chujm@mails.jlu.edu.cn (J.C.); chenwq1313@mails.jlu.edu.cn (W.C.); qzzhao20@mails.jlu.edu.cn (Q.Z.); caicy1322@mails.jlu.edu.cn (C.C.); 2Henan International Joint Laboratory for Nuclear Protein Regulation, School of Basic Medical Sciences, Henan University, Kaifeng 475004, China; saadullah@henu.edu.cn

**Keywords:** cancer, microRNA, miR-372-3p, INO80 chromatin remodeling complex, transcriptional regulation

## Abstract

Human INO80 chromatin remodeling complex (INO80 complex) as a transcription cofactor is widely involved in gene transcription regulation and is frequently highly expressed in tumor cells. However, few reports exist on the mutual regulatory mechanism between INO80 complex and non-coding microRNAs. Herein, we showed evidence that the INO80 complex transcriptionally controls microRNA-372 (miR-372) expression through RNA-Seq analysis and a series of biological experiments. Knocking down multiple subunits in the INO80 complex, including the *INO80* catalytic subunit, *YY1*, *Ies2*, and *Arp8*, can significantly increase the expression level of miR-372. Interestingly, mimicking miR-372 expression in HCT116 cells, in turn, post-transcriptionally suppressed INO80 and Arp8 expression at both mRNA and protein levels, indicating the existence of a mutual regulatory mechanism between the INO80 complex and miR-372. The target relationship between miR-372 and INO80 complex was verified using luciferase assays in HCT116 colon cancer cells. As expected, miR-372 mimics significantly suppressed the luciferase activity of pMIR-luc/INO80 and pMIR-luc/Arp8 3′-UTR in cells. In contrast, the miR-372 target sites in the 3′-UTRs linked to the luciferase reporter were mutagenized, and both mutant sites lost their response to miR-372. Furthermore, the mutual modulation between the INO80 complex and miR-372 was involved in cell proliferation and the p53/p21 signaling pathway, suggesting the synergistic anti-tumor role of the INO80 complex and miR372. Our results will provide a solid theoretical basis for exploring miR-372 as a biological marker of tumorigenesis.

## 1. Introduction

Colorectal cancer (CRC) is the third most common cancer diagnosed globally, with over 1.84 million new cases reported in 2018, preceding only malignancies of the lung and breast [1]. Nowadays, CRC is the fourth leading cause of cancer death, accounting for 9.2% of deaths worldwide and posing a severe public health problem [2]. Although the 5-year survival rate of CRC patients has exceeded 65%, the treatment of CRC has become more complex and challenging because of its wide range of risk factors, including environmental factors, family history, age, obesity, smoking, alcohol use, lack of physical exercise, and poor nutrition [2,3,4]. Histological characteristics have been the only indication of a patient’s prognosis for colon adenocarcinoma (COAD) until now. Therefore, identifying the factors affecting the growth of CRC and the molecular processes involved in CRC progression is an important part of the treatment of CRC and is also the key to discovering biomarkers for COAD prognosis.

MicroRNAs (miRNAs) are 19–22 nucleotide non-coding single-stranded RNAs in plants, animals, and viruses [5]. It can post-transcriptionally regulate gene expression by controlling translation or mRNA degradation [6]. As the major regulators of gene expression, miRNAs have been discovered to control many important biological processes, such as cell proliferation, apoptosis, and migration [7]. Furthermore, numerous studies have shown that the imbalanced expression of miRNAs may result in the aberrant expression of target proteins, which can affect numerous biological processes in cells. According to this viewpoint, abnormal miRNA expression has been seen in a variety of malignancies, including gastric [8], lung [9], and prostate cancer (PCa) [10], indicating that miRNAs play a crucial role in the development of cancer. MiRNA transcriptional silencing is a well-known mechanism of epigenetic alterations in cancer. Among the silenced miRNAs, miR-372 and miR-373 have been linked to DNA hypermethylation. However, it is worth mentioning that histone modifications may also potentially influence their expression. Inhibitors of histone deacetylase (HDAC), such as suberoylanilide hydroxamic acid (SAHA) and trichostatin A (TSA), can restore the expression of miR-373 in LNSC A549 and Calu-6 cells. This restoration inhibits cell proliferation and invasion by decreasing the expression of miR-373 target genes, interleukin-1 receptor-associated kinase 2 (*IRAK2*), and lysosomal-associated membrane protein 1 (*LAMP1*) [11]. On the other hand, miR-520 and miR-373 are responsible for the upregulation of matrix metalloprotein 9 (*MMP9*) in human fibrosarcoma HT1080 cells by directly targeting the 3′-UTR of mTOR and sirtuin 1 (*SIRT1*) mRNAs. This upregulation activates the Ras/Raf/MEK/Erk signaling pathway and NF-B, increasing cell growth [12].

According to the literature, human embryonic stem cells (ESCs) contain microRNA-371–373 (miR-371–373) gene clusters in a highly abundant form, indicating that miR-371–373 genes are important in the maintenance of stemness in ESCs [13]. Through this, the miR-371–373 gene cluster, involved in signaling pathways, such as the Wnt/β-catenin pathway, can promote stem cell self-renewal and oncogenesis in various tissues [14]. For instance, miR-372 and miR-373 promote the stemness of CRC cells by suppressing the expression of differentiation genes, such as the nuclear factor kappa B (*NF-kB*), mitogen-activated protein kinase (*MAPK*)/extracellular signal-regulated kinase (*Erk*), and vitamin D receptor (*VDR*) [15]. Accordingly, several primarily diagnosed cancer types have been reported to express miR-371–373 abnormally. It is important to note that miR-371–373 expression levels differ depending on the kind of cancer. For example, the expression of miR-372–373 is found to be significantly reduced in non-small cell lung cancer [11], pancreatic cancer [16], prostate cancer [17], cervical cancer tissues [18], endometrial cancer [19], and ovarian carcinoma [20] as compared to normal tissues. On the other hand, miR-372 was found overexpressed in several tumor tissues compared to normal tissues, including esophageal squamous cell carcinoma [13], lung squamous cell carcinoma [21], oral squamous cell carcinoma [22], hepatocellular carcinoma [23], and glioma [24]. It is worth noting that several studies have reported contradictory results for miR-372 expression in breast cancer tissues and CRC: reduced miR-372 expression in 20 and 45 primary breast cancer and CRC tissues compared to paired normal tissues was reported [25,26], respectively, whereas increased miR-372 expression in the same type of cancer tissues was also reported [27,28], highlighting the complexity of the molecular mechanism of miRNAs in tumorigenesis. In a word, the expression of miRNAs and the genes they regulate are strongly linked to cancer.

Growing evidence shows that miR-372 regulates many important biological processes in cells, such as cell proliferation, apoptosis, migration, and invasion, in many human cancers [29]. MiR-372 may operate as tumor suppressors by inhibiting oncogene expression or as oncogenic miRNAs by repressing the expression of tumor suppressor genes. For example, miR-372 promotes the proliferation of CRC, breast cancer, and gastric cancer cells by binding to target sites in the 3′-UTR of large tumor suppressor homolog 2 (*LATS2*) [30]. In addition, miR-372 plays an oncogenic function by down-regulating fibroblast growth factor 9 (*FGF9*) and *p62*, respectively, in both lung squamous and head and neck squamous cell carcinoma [21,31]. On the contrary, miR-372 functions as a tumor suppressor by targeting and binding to complementary sequences in the 3′-UTRs of genes, such as FXYD domain-containing ion transport regulator 6 (*FXYD6*), inhibiting growth and metastasis in different cancer cells, including osteosarcoma, renal cell carcinoma, prostate cancer, hepatocellular carcinoma, breast cancer, and cervical cancer cells [32]. However, there are few reports on the relationship between miR-372 expression or its function and epigenetic regulation.

The human INO80 chromatin remodeling complex composed of 15 subunits catalyzes ATP-dependent sliding of nucleosomes along DNA. Its eight core subunits (Arp5, Arp8, TIP49a/b, Ies2, Ies6, Arp4, and YY1) are highly evolutionarily conserved from *yeast* to humans and form an enzymatic core, including HSA (helicase-SANT-associated domain) and SNF2 (SWI/SNF catalytic subunit 2) module [33,34]. Therefore, dysfunction of any of the catalytic subunits of INO80 or the enzyme core may affect the biological function of the entire complex of INO80 [35,36]. Based on gene expression profiles from the knockdown of *INO80*-, *Arp8*-, *Arp5*-, *Ies6*-, and *Ies2* in HeLa cells, we found that hundreds of genes were co-regulated by silencing the SNF2 (Arp5, Ies6, and Ies2) and HSA (Arp8) modules. Among the gene expression profiles, upregulation of miR372 was found in INO80 complex key subunits knockdown gene profiles. In line with this, two YY1-binding sites were identified upstream of the miR-372 transcription start site, and upregulation of miR-372 was found in siYY1 treated MCF-7 breast cancer cells (YY1 is known to be one of the key subunits of the INO80 complex) [37]. Therefore, we speculate a tight relationship between the INO80 complex and miR-372. Given that the INO80 complex has a wide range of functions in cells, such as transcriptional regulation, genome stability, nucleosome remodeling activity, and tumorigenesis [35,36], studies of the mutual regulation between the INO80 complex and miR-372 are of great significance for elucidating the mechanism of INO80 complex in cells. Thus, this study aims to clarify the transcriptional regulation of INO80 on miR-372 and the possible feedback mechanism between miR-372 and the INO80 complex. Our study will provide a solid theoretical basis for exploring miR-372 as a biological marker of tumorigenesis.

## 2. Results

### 2.1. DNA Microarray Analysis of Gene Expression from HeLa Cells Indicated That INO80 Complex Mainly Regulates Cell Proliferation and Cell Viability

The INO80 complex contains six metazoan-specific subunits, which all assemble on the N-terminus of the INO80 protein and form an N-terminal regulatory module (Figure 1A). To investigate the target genes of the INO80 complex, subunits of the INO80 complex, including *INO80*, *Arp5*, *Arp8*, *Ies2*, and *Ies6*, were knocked down with specific siRNAs in HeLa cells. The total RNA from HeLa cells was extracted, and the knockdown efficiency was measured by RT-qPCR (Figure 1B). After that, the total RNA samples were sent for DNA microarray analysis. The Illumina microarray datasets are accessible from the National Center for Biotechnology Information Gene Expression Omnibus (GEO) data repository (accession#: GSE68655). As shown in Figure 1C, 9478 genes were differentially expressed among INO80, Arp8, Arp5, hIes6, or hIes2 and non-targeting (NT) siRNA knockdown HeLa cells. To further identify the function of genes regulated by INO80, Arp8, Arp5, Ies2, and Ies6, enriched Gene Ontology (GO) terms were analyzed, as shown in Figure 1D. GO terms enriched by INO80-, Arp8-, Arp5-, Ies2-, and Ies6 represented housekeeping functions that were related to cell proliferation, cell growth, cell adhesion, cell migration, cell motion, cell division, and cell motility, suggesting the important roles of INO80 complex in regulating these basic cell biological functions. In addition, hundreds of overlapping genes were found to be regulated by INO80, Arp8, Arp5, hIes6, and hIes2, which are components of the HSA and SNF2 modules (accession#: GSE68655).

### 2.2. The Silencing of INO80 Caused HeLa Cells to Grow in Clusters and Inhibited the Colony-Forming Ability of HCT116 Cells

To further confirm the reliability of DNA microarray analysis results, we decided to establish INO80-inducible silencing cell lines. Using the pSingle-tTS-shRNA system, we first constructed the pSingle-tTS-shINO80 plasmid, then the HeLa cells were transfected with pSingle-tTS-shINO80 plasmids and selected using neomycin to obtain *INO80*-inducible knockdown HeLa cells. As shown in Figure 2A, when the selected pSingle-tTS-shINO80 cells were treated with doxycycline (Dox), it was observed that the level of INO80 protein decreased gradually with the increase in Dox concentration, which proved that Dox could induce the silencing of *INO80* gene in cells. Moreover, to observe the effect of *INO80* silencing on cell growth status, 2 μg/mL Dox was used to induce INO80 silencing for 2 weeks. Compared with cells before Dox-induction, silencing of *INO80* by Dox caused the cells to grow in clusters and resulted in changes in cell morphology. This phenomenon became obvious over time (comparing the cells at 1 and 2 weeks), suggesting the involvement of the INO80 complex in regulating the normal cell growth process (Figure 2B). In addition, using the pLVX-shRNA lentivirus knockdown system, we established a stable knockdown *INO80* cell line in HCT116 colorectal cancer cells. Stably expressing pLVX-shINO80-1 and pLVX-shINO80-2 cells were confirmed by western blotting (Figure 2C). To further verify whether the knockdown of *INO80* affects cell proliferation of the HCT116 cells, the colony formation assay was performed using pLVX-shNT- and pLVX-shINO80- plasmids transfected HCT116 cells. As shown in Figure 2D, downregulation of *INO80* significantly inhibited the proliferation of HCT116 cells (upper), and the quantified number of colonies in each group was shown in (lower) (*** *p* < 0.001). Additionally, fluorescence-activated cell sorting (FACS) analysis was performed for pLVX-shNT and pLVX-shINO80 transfected HCT116 cells and observed cell cycle arrest at the G2/M phase as compared to the control empty vector, as shown in Figure 2E.

### 2.3. Elevated pri-miR-372 Expression Level Was Detected in INO80-Complex Knockdown HCT116 Cells

As mentioned in Figure 1C, 9478 genes were differentially expressed in INO80 complex knockdown HeLa cells. Hundreds of overlapping genes were found to be co-regulated by INO80, Arp8, Arp5, hIes6, and hIes2. Interestingly, miR-372 and miR-373 also exist in co-regulated genes (Figure 3A), suggesting the INO80 complex may regulate the expression of miR-372/miR-373. Therefore, to address this speculation, we designed RT-PCR primers to detect *miR-372* expression. At first, HCT116 cells were transiently transfected with 20 pmol of siNT, siIes2, and siArp8 for 48 h. After confirming the knockdown efficiency (Figure 3B), pri-*miR-372* levels were then measured by RT-qPCR. As a result, pri-*miR-372* levels were significantly increased by knocking down *Ies2* (* *p* < 0.05) and *Arp8* (*** *p* < 0.001) (Figure 3C). Next, HCT116 cells were transiently transfected with 20 pmol of non-targeting siRNA (siNT) and 10 or 20 pmol of siYY1; 48 h later, cells were harvested, and the whole cell lysate and total RNAs were prepared. The knockdown efficiency of YY1 was detected through western blot (Figure 3D), and pri-*miR-372* was significantly upregulated by knocking down *YY1* in a dose-dependent manner (Figure 3E). Similar results were obtained in pLVX-shYY1-1 or pLVX-shYY1-2 transfected HCT116 cells. As shown in Figure 3F, the knockdown efficiency of YY1 was measured using the western blot method, and a significant increase in pri-*miR-372* level in *YY1* knocked down HCT116 cells was observed (Figure 3G). Finally, we detected pri-*miR-372* level in *INO80* knockdown HCT116 cells. To silence INO80, HCT116 cells were transfected with pSingle-tTS-shINO80 plasmids with or without 2 μg/mL Dox treatment. The experiment design is shown in Figure 3H—Dox-induced reduction of INO80 protein levels. Simultaneously, the pri-*miR-372* levels were increased in a dose-dependent manner (Figure 3I). In conclusion, the elevated *miR-372* expression was observed by knocking down multiple subunits of the INO80 complex, confirming that the INO80 complex regulates *miR-372* transcription.

### 2.4. Mimics miR-372 in HCT116 cells, in Turn, Suppressed INO80 and Arp8 Expression in Both mRNA and Protein

The above results suggest that the INO80 complex can transcriptionally regulate the expression of miR-372. To further clarify the mutual function between the INO80 complex and miR-372, we searched the possible targeted regulatory genes of miR-372 using the Target Scan Human (https://www.targetscan.org, accessed on 5 March 2021) website. As a result, the complementary sequences of miR-372 were found in the 3′-UTR of the multiple subunits of the INO80 complex, including INO80, YY1, INO80D, Arp8, and MCRS1 (Table 1), suggesting that the expression of *miR-372* is not only transcriptionally regulated by INO80 complex, but in turn, miR-372 may affect the expression level of INO80 complex by binding to the 3′-UTR of the above genes; this may further affect the stability and biological function of the INO80 complex. We constructed the pmR-mCherry-miR-372 plasmid to prove this assumption, which can express miR-372 in mammalian cells. To confirm *miR-372* expression, HCT116 cells were transfected with pmR-mCherry empty vector and pmR-mCherry-miR-372 plasmids for 48 h, cells were then collected, and the total RNA was extracted. The pri-*miR-372* levels were analyzed by RT-qPCR using specific primers. Compared with the empty vector group, a significant dose-dependent increase of *miR-372* was observed (*** *p* < 0.001) (Figure 4A), and this expression was effectively inhibited by co-transfecting with miR-372-3p inhibitors (Figure 4B). Thus, we examined the effect of miR-372 on the expression level of key subunits of the INO80 complex. As shown in Figure 4C, after transfection of pmR-mCherry-miR-372 plasmids for 48 h in HCT116 cells, INO80 and Arp8 protein levels were decreased in mimic *miR-372* expression cells. In addition, the relative mRNA levels of INO80 and Arp8 in HCT116 cells were significantly decreased by the expression of *miR-372* dose-dependent (Figure 4D–E). By examining multiple vector concentrations, we aimed to determine the optimal dosage that would yield the most significant and consistent effects of miR-372 on our target. In the colony formation assay, the clonogenic ability of HCT116 cells was suppressed by transfecting with miR-372-3p, and this effect was antagonized by inhibitors (Figure 4F upper). The quantified number of colonies in each group is shown in (Figure 4F lower). This result is consistent with inhibiting colony-forming ability by knocking down the *INO80* gene. Combined with the previous experimental results, undoubtedly, miR-372 regulates the INO80 and Arp8 expression levels in both mRNA and proteins in HCT116 cells. Therefore, it can be speculated that miR-372 may regulate the function of the INO80 complex by targeting multi-subunits in the complex.

### 2.5. MiR-372 Post-Transcriptionally Inhibited INO80/Arp8 Expression by Targeting Their 3′-UTR in HCT116 Cells

The previous experimental results suggest that Arp8 and INO80 may be miR-372 targeted regulatory genes. To confirm whether miR-372 can bind to the 3′-UTR of INO80 and Arp8 and post-transcriptionally regulate their expression levels, we predicted possible binding sites in the 3′-UTR of *INO80* and *Arp8*. We found two possible binding sites in the 3′-UTR of *INO80* (153–159 and 4644–4650) and *Arp8* (457–463 and 1594–1601), respectively, and introduced the DNA fragments containing the binding sites into the pMIR-REPOIRT-Luc vector to construct the luciferase reporter gene plasmid. Concurrently, the mutagenic plasmid of the miR-372 binding site in the INO80-3′-UTR was designed and constructed (Figure 5A). First, pMIR-REPORT-INO80-3′-UTR and pMIR-REPORT-Arp8-3′-UTR plasmids (a vector carrying the wild-type 3′-UTR of *Arp8* and *INO80*) were transiently transfected into HCT116 cells, and the luciferase activities were detected in the presence or absence of miR-372. As shown in Figure 5B, the luciferase activities were significantly inhibited by co-transfection with miR-372 compared to the no-miR-372 group (** *p* < 0.01, *** *p* < 0.001). As expected, co-transfection of pMIR-REPORT-INO80-3′-UTR or pMIR-REPORT-Arp8-3′-UTR with miR-372 reduced Arp8 (Figure 5C, lanes 2 and 4) or INO80 (Figure 5D, lanes 3 and 4) protein levels compared those to the transfection of pMIR-REPORT-INO80-3′-UTR or pMIR-REPORT-Arp8-3′-UTR only group, suggesting that INO80 and Arp8 may be the targets of miR-372. Furthermore, while the miR-372 target sites in the 3′-UTR of INO80 linked to the luciferase reporter were mutagenized, both mutant sites lost their response to miR-372, and the INO80 protein level no longer decreased due to the co-transfection of miR-372 (Figure 5E, lanes 5 and 6), indicating the site-specificity of the repression. The above experimental results suggest that there is a functional interaction between the INO80 complex and miR-372: INO80 regulates the expression of miR-372, which in turn affects the stability and function of the INO80 complex by binding to its 3′-UTR (Figure 5F).

### 2.6. Mimics miR-372 Upregulated the p53/p21 Pathway in HCT116 Cells

CDKN1A (p21) is a universal inhibitor of cyclin kinases that controls the cell cycle by activating and inactivating the cyclin-dependent kinases (CDKs) [38]. In addition, p21 carries out its diverse biological functions by detecting and responding to multiple signals through the p53-dependent pathway and p53-independent pathway. Its primary role as a tumor suppressor is attributed to its ability to impede cell cycle progression and facilitate DNA damage repair. Nevertheless, considerable evidence suggests that p21 can also exhibit oncogenic properties primarily by suppressing apoptosis [39]. We previously demonstrated that the INO80 complex negatively regulates the *p21* expression [35]. Based on the published literature, there are two binding sites (−2.5–2.2 kb and −1.3–1.0 kb) upstream of the *p21* transcriptional start site (TSS) for the tumor suppressor gene *p53*, which is a regulator of p21 [40]. Thus, to understand the enrichment of INO80 near the *p21* transcription start site (TSS) as the cell cycle changes, we designed six pairs of qPCR primers, including two p53 binding sites, in the *p21* locus to amplify ChIP DNA (Table 2) (Figure 6A). First, HCT116 cells were synchronized with 1 mM hydroxyurea (HU) at the G1/S phase. Then, cells were released from G1/S arrest by HU. We harvested cells at different times to collect G1, S, and G2/M cells (Figure 6B, upper). ChIP assays were then performed using INO80 antibody in different cell cycle phases HCT116 cells. As expected, INO80 occupied the p53 binding sites (−2.2 kb and −1.0 kb upstream of the p21 TSS) at G1 and S phases but not at the G2/M phase (Figure 6B, lower). It suggests that INO80 may activate *p21* transcriptional expression through p53. In line with this prediction, after knocking down *INO80* with Dox induction, increased p53 and p21 protein levels were observed. In addition, the knockdown of *INO80* resulted in decreased Arp8 protein in cells (Figure 6C). Interestingly, HCT116 cells that mimic *miR-372* expression not only caused a reduction of INO80 protein expression but also led to the increase of p53 and p21 protein levels (Figure 6D), suggesting that miR-372 may post-transcriptionally inhibit the expression of *INO80* through binding to INO80-3′-UTR, thereby reducing the expression of INO80 targeted gene *p21*. Our subsequent experimental results also support this assumption. In Dox-induced *INO80* knockdown HCT116 cells, mimic *miR-372* significantly upregulated the p21 and p53 protein levels (lane 3) compared to those in *INO80* knockdown-only group (lane 2) (Figure 6E). The above data suggest that miR-372 may post-transcriptionally regulate the stability of the INO80 complex and play roles through co-regulating the p53/p21 pathway in cells.

## 3. Discussion

In this report, we first proposed that the INO80 complex and miR-372 are mutually regulated in HCT116 colorectal cancer cells using in vitro biological experiments combining knockdown and over-expression approaches. In a transcriptome sequencing analysis from our previous study, we found that the INO80 complex upregulated miR-372 transcription in HeLa cells. The possible reason is that the INO80 complex, as a chromatin remodeling complex, is widely involved in the transcriptional regulation of multiple genes by altering chromatin structure. Upon knocking down the INO80 complex in different types of cancer cells, the proteins (or complexes) or transcription cofactors recruited by INO80 in the promoter region of the miR-371–373 gene cluster may have different effects on the expression level of miR-372–373. Furthermore, according to the currently reported literature, the expression level of miR-372 varies depending on the type of cancer. For example, compared to normal tissues, miR-372–373 is significantly reduced in tissues such as non-small cell lung cancer [11], pancreatic cancer [16], cervical cancer [18], endometrial cancer [19], and ovarian cancer [20].

On the contrary, the expression of miR-372 is higher in lung squamous cell carcinoma [21], colorectal cancer [15], esophageal squamous cell carcinoma [13], and hepatocellular carcinoma [23] than in normal tissues. It is worth mentioning that in certain specific cancer types, the expression level of miR-372 contradicts the results reported by different research groups, such as breast cancer and colon cancer [25,26]. This influence pattern was further confirmed in HCT116 cells by knocking down multi-subunits of the INO80 complex, including INO80, Arp8, Ies2, and YY1. On the other hand, by combining methods of mimic miR-372 expression and dual luciferase assays, we postulate that the role of the miR-372 in post-transcriptionally regulating the expression of the INO80 complex by targeting and binding to the complementary sequences in the 3′-UTR of INO80 and Arp8. In addition, we speculate that the INO80 complex and miR-372 may be involved in the p21/p53 pathway in a coordinated regulation mode.

Evolutionarily conservative shared subunits of the human INO80 complex assemble on conserved ATPase and HSA domains of the INO80 catalytic protein to maintain nucleosome sliding and ATPase activity [33,34]. Any subunit, including INO80 catalytic protein, Arp8, YY1, Ies2, Arp5, Ies6, and Tip49a/b, deficiency in the HSA and SNF2 modules may lead to the loss of function of INO80 complex, which in turn affects transcriptional gene regulation [41]. In line with this, a total of 9478 genes were differentially expressed due to knocking down *INO80*, *Arp8*, *Arp5*, *Ies6*, and *Ies2*, which are tightly related to cell proliferation, cell growth, cell adhesion, cell migration, cell motion, cell division, and cell motility, suggesting the importance of the INO80 complex in regulating basic cell biological functions. It is worth noting that miRNAs are also regarded as targets of epigenetic regulators such as DNA methyltransferases (DNMTs) and histone deacetylases (HDACs) [42]. Consistent with this view, knocking down multi-subunits (*INO80*, *Arp8*, *YY1*, or *Ies2*) of the INO80 complex upregulated pri-*miR-372* in HCT116 cells. In addition, miR-372 was also identified as one of the targets from INO80 complex knockdown of gene expression profiles. The above results suggest that miR-372 is a potential target of the INO80 complex. Our findings are directly in line with previous findings reported by Feng and his colleagues [37], that the upstream region of the miR-372 contains two YY1 consensus binding sites (−1331 and −159 sites), suggesting that YY1 may directly control *miR-372* transcriptional regulation. YY1 is a member of the INO80 complex and a DNA-binding protein. YY1 can initiate, activate, or repress cell gene transcription by recruiting different transcriptional cofactors to its activation or repression domain [43]. For example, YY1 transactivates *CDC6* and *GRP78* by recruiting the INO80 complex to the DNA binding sites on their promoters [44]. INO80 complex is well-known for its capability to regulate many target genes via recruitment at specific genome sites and remodeling the chromatin structure. Our previous study has also revealed the direct influence of INO80 on the p53/p21 pathway by prompting the removal of H2A.Z at the p53-binding site of the p21 gene in response to doxorubicin [45]. Therefore, we speculate that the INO80 complex may regulate the transcription *miR-372* by binding to its promoter through YY1 and regulating the local chromatin structure, as we have previously revealed in the INO80- p21 regulatory axis. However, further studies are needed to elucidate the association between INO80-mediated chromatin remodeling and miR-372 expression in cancer cells.

On the other hand, our results strongly suggest that miR-372 post-transcriptionally regulates the expression of the INO80 complex. According to the Target Scan Human (https://www.targetscan.org, accessed on 5 March 2021) website analysis, complementary sequences of miR-372 in the 3′-UTR of multi-subunits of INO80 complex, including *INO80*, *Arp8*, *YY1*, *INO80D*, and *MCRS1*, were predicted. In our experiments, *miR-372* mimics reduced the expression levels of INO80 and Arp8 in both mRNA and protein. Moreover, mimic *miR-372* expression significantly suppressed the luciferase activities of pMIR-luc/INO80-3′-UTR and pMIR-luc/Arp8-3′-UTR in HCT116 cells, suggesting that miR-372 modulates the expression of *INO80* and *Arp8* by directly targeting their 3′-UTR. Based on the published literature, miR-372 can act as a tumor suppressor or an oncogenic factor depending on the gene targeted in different tumor cells. For instance, overexpression of *miR-372* promotes the proliferation and migration of colorectal cancer cells by inhibiting the tumor suppressor gene *LATS2* [30]. In our experiments, either *miR-372* mimics or *INO80* silencing markedly suppressed the clonogenic ability of HCT116 cells, supposing that the miR-372 inhibiting colony formation is at least partially caused by inhibition of the INO80 complex. Through this, *INO80* silencing selectively inhibits melanoma cell proliferation, tumorigenesis, and tumor maintenance in mouse xenografts [46], indicating that miR-372-INO80 complex regulatory axis may function as a tumor suppressor in cells. Combined with our experimental results and previously reported data, the INO80 complex not only occupies p53-binding sites of the *p21* promoter but also negatively regulates p21 expression in a p53-dependent manner, further affecting the cell cycle process and maintenance of chromosome stability [35]. In HCT116 cells, mimicking *miR-372* expression or knockdown of *INO80* could upregulate p53 and p21 protein levels. Moreover, in Dox-induced INO80 knockdown HCT116 cells, *miR-372* mimics significantly upregulated the p21 and p53 protein levels compared to INO80 knockdown only group, suggesting that the INO80 complex and miR-372 may be involved in the p21/p53 pathway in a coordinated regulation mode.

In conclusion, the experimental results in this paper reveal the mutual regulatory relationship between miR-372 and the INO80 complex and make it clear that miR-372 can directly interact with the 3′-UTR of multiple components in the complex, thereby post-transcriptionally regulating the expression and stability of the complex, altering the complex’s chromatin remodeling activity, which in turn further affects the intracellular biological function of the INO80 complex. As shown in the schematic diagram below (Figure 7), in normal cells, both miR-372 and p53/p21 are suppressed by the INO80 complex by regulating the chromatin structure; when the INO80 complex is silenced, it will lead to instability of the complex and loss of chromatin remodeling enzyme activity; thus, the inhibition of INO80 complex on miR-372 and p53/p21 is released, leading to the up-regulation of miR-372 and p53/p21 expression, further inhibiting the proliferation of HCT116 colon cancer cells. However, most data supporting the above mutual regulation model are limited to HCT116 cells in this study, and whether such a regulatory pattern also exists in other cancer types still needs to be verified. In summary, the results of this paper provide a theoretical basis for elucidating the mechanism of action of the INO80 chromatin complex and provide new ideas for the development of subsequent cancer therapeutics. MiR-372 is also expected to serve as a biomarker for early cancer diagnosis.

## 4. Materials and Methods

### 4.1. Antibodies

Anti-INO80 (24819-1-AP), anti-BCCIP (16043-1-AP), and anti-p21 (10355-1-AP) antibodies were purchased from Proteintech Group (Wuhan, China). Anti-YY1 (H414; sc-1703) and anti-α-Tubulin antibodies were obtained from Santa Cruz Biotechnology (Dallas, TX, USA). Anti-p53 mouse monoclonal antibody was provided by Boster Group (BM0101, Wuhan, China). Anti-Arp8 and anti-GAPDH were raised against bacterially expressed proteins (Jilin University).

### 4.2. Cell Culture

HCT116 colon cancer and HeLa cervical cancer cell lines were obtained from the Type Culture Collection of the Chinese Academy of Sciences (Shanghai, China), and all experiments were performed with mycoplasma-free cells. Cells were cultured in Dulbecco’s modified Eagle’s medium (DMEM, Gibco, Life Technologies, Waltham, MA, USA) medium containing 10% fetal bovine serum (KangYuan Biology, Tianjin, China) and 1% penicillin-streptomycin (Thermo-Fisher Scientific, Waltham, MA, USA) and maintained at 37 °C and 5% CO_2_.

### 4.3. Reverse Transcription PCR

Total RNA was extracted from HeLa and HCT116 cells using RNAiso Plus (9109, Takara, Tokyo, Japan). cDNA synthesis was performed with a PrimeScript First Strand Kit (6110A, Takara, Tokyo, Japan) and a specific primer to elongate miR-372-3p. Relative mRNA levels of miR-372-3p, INO80, Arp8, YY1, Ies2, U6, and GAPDH were evaluated using real-time qPCR with the SYBR^®^Premix EX Taq^TM^II kit (RR820A, Takara, Tokyo, Japan). Initial denaturation at 95 °C for three minutes was followed by 35 cycles of denaturation at 95 °C for 30 s, annealing at 60 °C for 30 s, and extension at 72 °C for 30 s. The primer sets used for RT-PCR are mentioned in Table 3. miR-372-3p expression was standardized to U6, while another mRNA expression was adjusted to GAPDH. There were three separate experiments with three replicates per group. Using the 2^−ΔΔCT^ approach, the relative expression levels of miR-372-3p, IN080, Arp8, and YY1 were determined.

### 4.4. Plasmids and Transient Transfection

A 341-bp DNA fragment containing pri-miR-372-3p was introduced between the pmR-mCherry vector’s XhoI and BamHI sites. Human INO80 3′-UTR fragments containing 153–159 and 4644–4650 sites and Arp8 3′-UTR fragments containing 457–463 and 1594–1601 sites were amplified and cloned using real-time PCR between the MulI and HindIII restriction enzyme sites, downstream of luciferase. Likewise, 3′-UTR mutants with mutated miR-372-3p binding sites INO80 were cloned into the pMIR-Reportluc vector between the same locations.

HCT116 cells were transfected with pmCherry-miR-372-3p, pmR-mCherry (as a negative control for pmCherry-miR-372-3p), or pMIR-Report-Luc/INO80-3′-UTR wild type (WT) or mutant (MT) and pMIR-Report-Luc/Arp8-3′-UTR WT or MT plasmids using polyethyleneimine (23966) (Polysciences, Shenzhen, China). Synthetic miR-372-3p inhibitors (AGAAUAGUGCUCCACAUUUGAGG) and miRNA inhibitor negative control (GenePharma, Shanghai, China) were transfected with Lipofectamine™ 2000 (Invitrogen, Carlsbad, CA, USA) following the manufacturer’s instructions. Cells were harvested after 48 or 72 h of transfection and lysed in buffer containing 1% NP-40, 150 mM NaCl, 50 mM Tris-HCl, 10% glycerol, 1 mM dithiothreitol, and complete protease inhibitor cocktails. Proteins in whole-cell lysates were analyzed by western blot using specific antibodies.

### 4.5. siRNA/shRNA Knockdown

Ies6-(Customized), Ies2-(Customized), INO80-(D-004176), and non-targeting (NT)-siRNAs (D-001206, as a control) siRNAs were ordered from Dharmacon (Lafayette, CO, USA). siYY1 (sc-36863), siArp8 (sc-60072) and siARP5 (sc-72442) were bought from Santa Cruz Biotechnology (Dallas, TX, USA). The suppliers did not provide the above siRNA sequences. The shRNA libraries were obtained from the RNAi Consortium (https://portals.broadinstitute.org/gpp/public/, accessed on 1 May 2021). Specific TRCN clone number for YY1 targeting was TRCN0000019894 (shYY1-1) and TRCN0000019897 (shYY1-2). Specific TRCN clone number for INO80 targeting was TRCN0000107555 (shINO80-1) and TRCN0000107558 (shINO80-2). pLVX vector without any shRNA insertion was used as sh-NT control. Total RNA or a whole-cell extract was collected 48 h after transfection for RT-PCR and western blot analysis.

### 4.6. Development of pSingle-tTS-INO80-shRNA Stable Cell Lines

HeLa cells were transfected with pSingle-tTS-shINO80 and pSingle-tTS-shArp8 plasmids and selected using neomycin to obtain INO80- and Arp8-inducible knockdown HeLa cells. Then, HeLa cells were treated with 2 μg/mL doxycycline (Dox) for 48 h. Post-transfection of 48 h, cells were collected for western blot and RT-PCR.

### 4.7. Luciferase Reporter Assay

HCT116 cells were co-transfected with 0.4 μg of pmCherry-miR-372-3p or the reporter mentioned above plasmids and encoded renilla and firefly luciferase vector as controls (0.12 ng). Twenty-four hours after transfection, the cells were collected. The luciferase activity of the pMIR-Report-Luc/INO80-3′-UTR and pMIR-Report-Luc/Arp8-3′-UTR activity was measured using firefly and renilla luciferase activities carried out using the Dual-luciferase Reporter assay kit (Promega, Madison, WI, USA), as well as by normalizing to renilla luciferase, as directed by the manufacturer. Three biological replicates were carried out.

### 4.8. Colony Formation Assay

HCT116 cells were treated with the control pLVX-vector, pLVX-shINO80, pmR-vector, and pmR-mCherry–miR-372-3p with or without miR-372 inhibitors. After 48 h of incubation, cells were digested using trypsin, re-suspended in Dulbecco’s modified eagle medium, and split into a new 12-well plate with 2 × 10^3^ cells/well. After 7 days of culture, formed colonies were stained with 0.1% crystal violet. Colonies with more than 20 cells were scored as positive. The Gel Imaging System (Liuyi Instrument Plant, Beijing, China) was used to take images of the colonies.

### 4.9. FACS Analysis

HCT116 cells were trypsinized after being treated with 0.5 mM HU for 0, 3, and 6 h. At 4 °C overnight, 10^6^ cells were suspended and fixed as single-cell dispersions in 70% ethanol. After washing two times with PBS, cells were resuspended in propidium iodide buffer (CF0031; Beijing Ding-guo, Beijing, China) and incubated for 30 min at 37 °C. EPICXLTM cytometers were used to obtain data (Beckman Coulter, Brea, CA, USA). MODFIT LT software (Version 5.0) was used to evaluate the collected data (Verity Software House, Topsham, ME, USA).

### 4.10. Chromatin Immunoprecipitation (ChIP)

HCT116 cells were cultured and grown to 80–90% confluence in a 10 cm plate, and ChIP assays were performed with the INO80 antibody according to a standard protocol. ChIP DNA was amplified with qPCR. The primer sets used for ChIP-PCR are mentioned in Figure 6A. Each experiment was performed 2–3 times. Antibodies and IgG-ChIP signals were normalized to total input.

### 4.11. DNA Microarray

As previously reported (34), specific genes, including hINO80, Arp5, Arp8, Ies2, and Ies6, were knocked down using siRNAs in HeLa cells, and the total RNA was sent for DNA microarray analysis to EMTD Science and Technology Development Co., Ltd. (Beijing, China). DAVID (Database for Annotation, Visualization, and Integrated Discovery), an online biological classification tool, was used to perform the enrichment analysis, Gene Ontology (GO) function analysis, and Kyoto Encyclopedia of Genes and Genomes (KEGG) pathway enrichment analysis of differentially expressed genes (DEGs) [47,48,49,50]. *p* < 0.05 and FDR < 0.02 genes utilized in the annotation were deemed statistically significant during the study. The Illumina microarray datasets are accessible from the National Center for Biotechnology Information Gene Expression Omnibus (GEO) data repository (http://www.ncbi.nlm.nih.gov/geo/) using the series accession number GSE68655.

### 4.12. Statistical Analysis

The data are reported as the mean ± standard deviation values. A two-tailed unpaired Student’s *t*-test was utilized to compare the two research groups statistically. Using SPSS version 19.0 (IBM Corporation, Armonk, NY, USA) or GraphPad Prism (GraphPad Software Version 5.0, San Diego, CA, USA), a one-way analysis of variance (ANOVA) followed by Tukey post-test was conducted to determine the differences between the three groups. *p* < 0.05 was considered statistically significant.

## Figures and Tables

**Figure 1 ijms-24-10685-f001:**
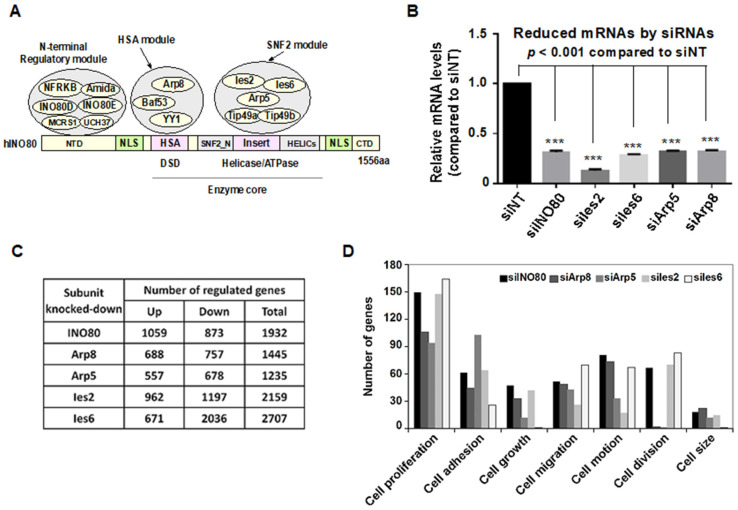
The INO80 complex was mainly involved in regulating cell proliferation and cell viability. (**A**) Functional modules of INO80 complex. NTD, N-terminal domain; HSA, helicase-SANT-associated domain; SNF2_N, SNF2 family N-terminal domain; Ins, insertion domain; HELICc, Helicase C-terminal domain; CTD, C-terminal domain. (**B**) Relative mRNA levels in siINO80, siArp8, siArp5, siIes2, and siIes6 transfected HeLa cells. *** *p* < 0.001 compared to the siNT group. (**C**) Differentially expressed gene numbers in INO80, Arp8, Arp5, Ies2, and Ies6 knockdown HeLa cells. (**D**) Gene Ontology (GO) terms analysis in INO80, Arp8, Arp5, Ies2, and Ies6 knockdown HeLa cells.

**Figure 2 ijms-24-10685-f002:**
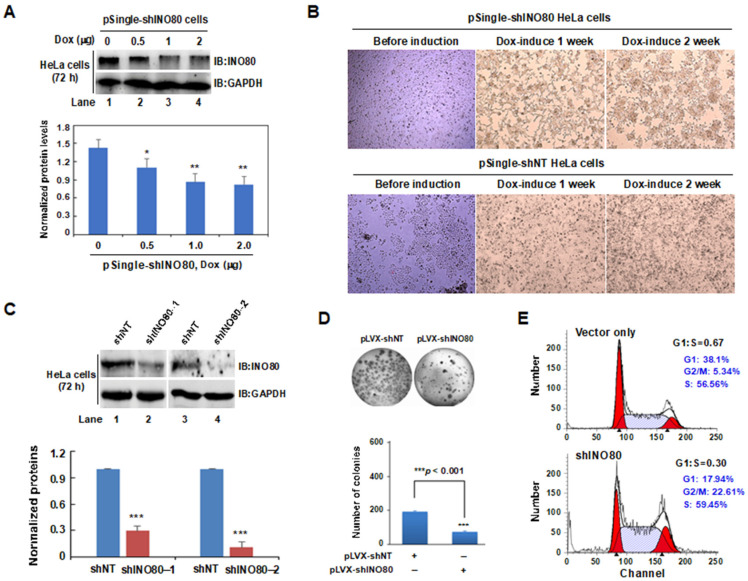
INO80 knockdown induced abnormal cell growth and proliferation in HeLa and HCT116 cells. (**A**) Dox-induced reduction of INO80 protein levels in HeLa cells. pSingle-tTS-shINO80 inducible HeLa cells were grown without or presence of Dox (0, 0.5, 1, and 2 µg/mL) for 72 h. GAPDH was used as an internal control (upper). INO80 protein levels were quantified by normalizing to the corresponding internal protein GAPDH using ImageJ software Version 1.8.0 (lower). * *p* < 0.05 or ** *p* < 0.01, compared to the no-Dox group. (**B**) Growth of wild-type (lower) and induced INO80 knockdown (upper) HeLa cells under an inverted microscope (magnification, ×200). (**C**) The western blot method confirmed decreased INO80 protein levels by pLVX-shINO80-1 and pLVX-shINO80-2. GAPDH was used as an internal control (upper). Normalized INO80 protein levels were shown in the lower panel. *** *p* < 0.001, compared to the shNT group. (**D**) Knockdown of INO80 inhibited HCT116 cell proliferation (upper) (magnification, ×200). Each group’s quantified numbers of colonies are displayed as a bar graph. *** *p* < 0.001, compared to the shNT group (lower). (**E**) Flow cytometry analysis of shINO80 knockdown HCT116 cell cycle.

**Figure 3 ijms-24-10685-f003:**
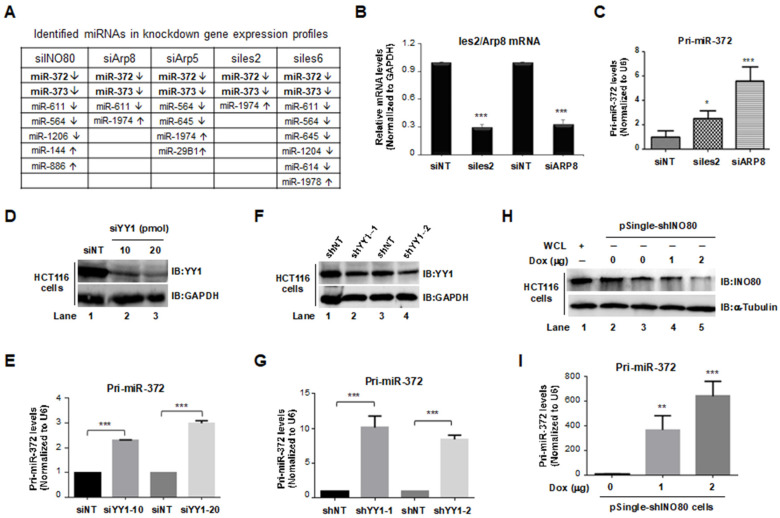
Elevated pri-*miR-372* expression level was detected in INO80-complex knockdown HCT116 cells. (**A**) miR-372/373 were identified in INO80 complex knockdown HeLa cells. Downregulation: ↓, Upregulation: ↑. (**B**) Knockdown efficiency of Ies2 and Arp8 in HCT116 cells. *** *p* < 0.001, compared to the siNT group. (**C**) Pri-*miR-372* expression levels in siIes2 or siArp8 transfected HCT116 cells. * *p* < 0.05 or *** *p* < 0.001, compared to the siNT group. (**D**) Knockdown efficiency of siYY1. GAPDH was used as an internal control. (**E**) Pri-*miR-372* expression levels in siYY1 transfected HCT116 cells. *** *p* < 0.001 compared to the siNT group. (**F**) The YY1 protein level in pLVX-shYY1-transfected HCT116 cells. GAPDH was used as an internal control. (**G**) Pri-*miR-372* expression levels in pLVX-shYY1 transfected HCT116 cells. *** *p* < 0.001, compared to the siNT group. (**H**) Dox-induced INO80 knockdown efficiency. α-Tubulin was used as an internal control. (**I**) Pri-*miR-372* expression levels in Dox-induced INO80 knockdown HCT116 cells. ** *p* < 0.01 or *** *p* < 0.001, compared to the no-Dox induction group.

**Figure 4 ijms-24-10685-f004:**
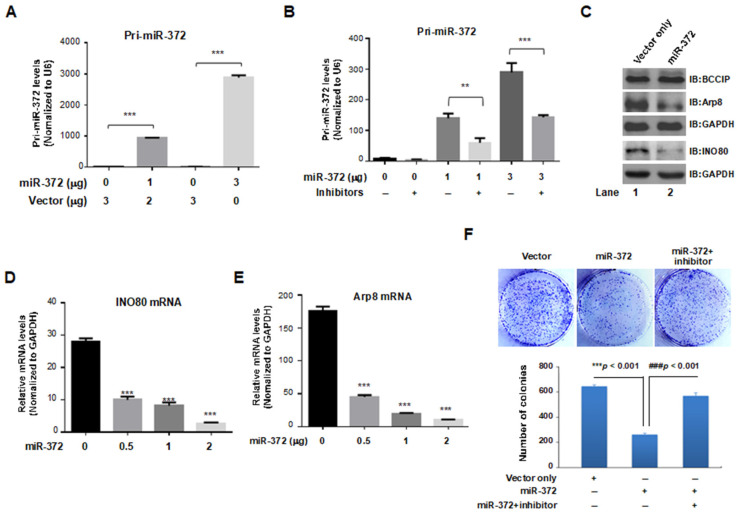
Pri-miR-372 suppressed the expression levels of INO80 and Arp8 in both mRNA and protein in HCT116 cells. (**A**) Validation of pmR-mCherry-miR-372 plasmid expression efficiency. *** *p* < 0.001, compared to vector only group. (**B**) The relative pri-*miR-372* levels in the presence or absence of pri-miR-372-inhibitors. ** *p* < 0.01, compared to the negative control group. (**C**) INO80 and Arp8 protein levels were estimated in miR-372 transfected HCT116 cells compared to the pcDNA3.1 group. The protein levels were analyzed by western blot with specific antibodies. Dose-dependent decrease of INO80 (**D**) and Arp8 (**E**) mRNA expression levels was observed in pmR-mCherry-miR-372 transfected HCT116 cells. (*** *p* < 0.01, compared to pcDNA3.1 group). (**F**) miR-372 mimics inhibited HCT116 cell proliferation. The cell proliferation abilities were estimated with colony-formation assays in HCT116 cells (upper) (magnification, ×200). Quantified numbers of colonies for each group are displayed as a bar graph (lower). Data are presented using mean ± standard deviation values. *** *p* < 0.001, vs. the vector groups, ^###^
*p* < 0.001, vs. the miR-372 group.

**Figure 5 ijms-24-10685-f005:**
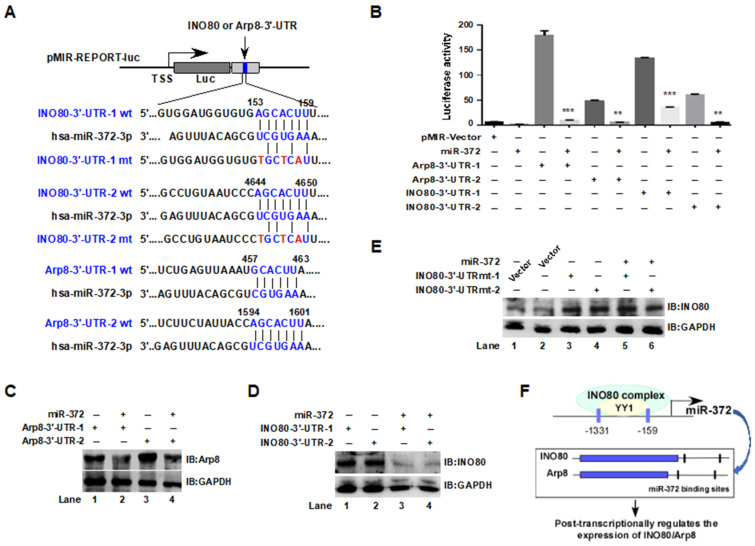
miR-372 post-transcriptionally regulated the expression of INO80/Arp8 by binding to their 3′-UTR in HCT116 cells. (**A**) Binding sites of miR-372-3p on the INO80 and Arp8 3′-UTR. The 3′-UTR fragments of human INO80 (wild type, wt; mutant, mt) and Arp8 (wt) were cloned downstream of the luciferase between the MluI and HindIII sites. (**B**) Relative luciferase activities. HCT116 cells were transfected with a plasmid containing pMIR-REPORT-Arp8-3′-UTR and pMIR-REPORT-INO80 3′-UTR with or without miR-372. ** *p* < 0.01, *** *p* < 0.001, vs. pMIR-REPORT-Arp8-3′-UTR or pMIR-REPORT-INO80 3′-UTR group. (**C**,**D**) Effects of miR-372 on the Arp8 and INO80 protein expression in HCT116 cells. Cells were co-transfected with pMIR-REPORT-Arp8-3′-UTR and pMIR-REPORT-INO80 3′-UTR plasmids with or without miR-372, and the protein levels were analyzed with a western blot. GAPDH was used as an internal control. (**E**) Mutant sites in pMIR-INO80-3′-UTR lost their response to miR-372. Lane 1 was transfected with an empty vector. (**F**) Possible working diagram.

**Figure 6 ijms-24-10685-f006:**
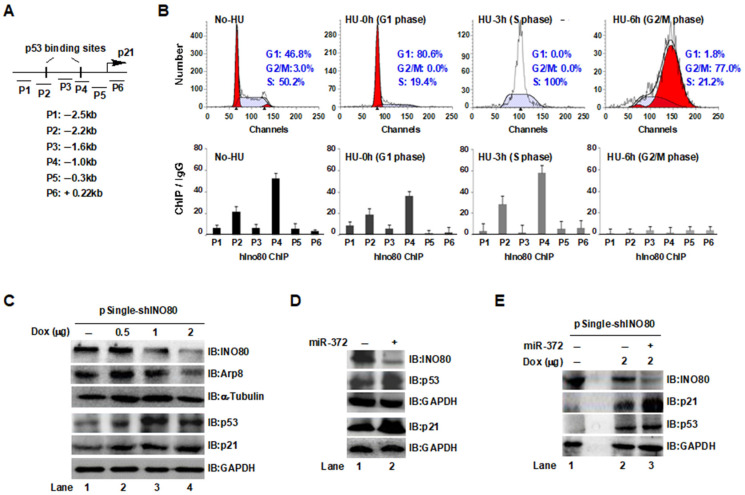
Mimic miR-372 expression upregulated the p53/p21 pathway in HCT116 cells. (**A**): Six primer sets in the p21 locus amplify ChIP DNA. (**B**): FACS and ChIP analyses. HCT116 cells were synchronized by treatment with 1 mM HU for 24 h of incubation. Cells were harvested by trypsinization 0, 3, and 6 h after removal of HU. Acquired data were analyzed using ModFit LT software, Version 5.0 (Verity Software House) (upper). ChIP assays were performed using INO80 antibodies in different cell cycle phase HCT116 cells. ChIP DNA was analyzed by qPCR. Bar graphs show the ratios of ChIP DNA signals to IgG (all signals normalized to input). Error bars represent the standard error of the mean of three independent experiments. (**C**): Detection of p21 and p53 proteins in Dox-induced INO80 knockdown HCT116 cells. (**D**): Effect of miR-372 on intracellular p21 and p53 protein levels. P21 and p53 proteins were detected in mimic miR-372 expression (48 h) HCT116 cells. (**E**): Interaction between miR-372 and INO80. After mimicking miR-372 expression in Dox-induced INO80 knockdown HCT116 cells for 48 h, p21, and p53 protein levels were analyzed with western blot.

**Figure 7 ijms-24-10685-f007:**
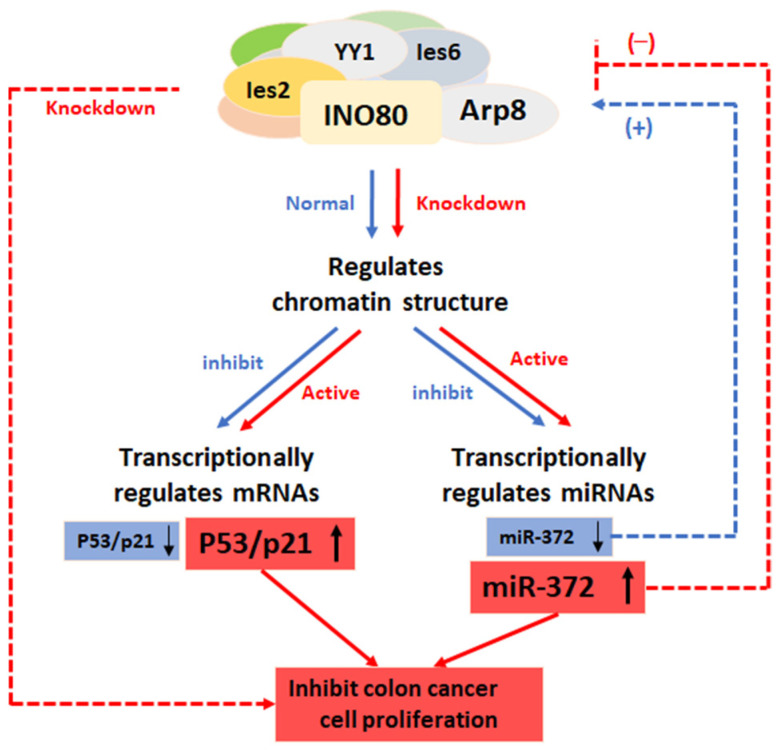
Working model of miR-372-INO80 complex axis. ↓, downregulation; ↑, upregulation.

**Table 1 ijms-24-10685-t001:** Complementary sequences of miR-372 were found in the 3′-UTR of the multi-subunits of the INO80 complex.

Target Gene	Representative Transcript	3P-seq tags + 5	Total Sites	7mer-m8 Sites	7mer-A1 Sites	Representative miRNA
INO80	ENST00000361937.3	279	2	1	1	has-miR-372-3p
YY1	ENST00000262238.4	1803	3	1	2	has-miR-372-5p
INO80D	ENST00000403263.1	5	2	1	1	has-miR-372-5p
ARP8	ENST00000335754.3	327	2	1	1	has-miR-372-5p
MCRS1	ENST00000546244.1	914	1	1	0	has-miR-372-5p

**Table 2 ijms-24-10685-t002:** Primer sets were used for ChIP-PCR on the promoter region of p21.

Primers	Directions	Sequences
−2.5 kb	forward	5′-ACATTGTTCCCAGCACTTCC-3′
	reverse	5′-TAGGGGAATGGTGAAAGGTG-3′
−2.2 kb	forward	5′-CTGTGGCTCTGATTGGCTTT-3′
	reverse	5′-CTCCTACCATCCCCTTCCTC-3′
−1.6 kb	forward	5′-TCTGGGGTTTAGCCACAATC-3′
	reverse	5′-CCTCTAACGCAGCTGACCTC-3′
−1.0 kb	forward	5′-TTGTCATTTTGGAGCCACAG-3′
	reverse	5′-GGGCTCAGAGAAGTCTGGTG-3′
−0.3 kb	forward	5′-GGGGCTCATTCTAACAGTGC-3′
	reverse	5′-GACACATTTCCCCACGAAGT-3′
+0.22 kb	forward	5′-CGTGTTCGCGGGTGTGT-3′
	reverse	5′-CATTCACCTGCCGCAGAAA-3′

**Table 3 ijms-24-10685-t003:** qRT-PCR primer sets used for PCR.

Primers	Directions	Sequences
hIno80	forward	5′-CGGAATCGGCTTTTGCTA-3′
	reverse	5′-TGTCGGCTGGTCAGTTGG-3′
hArp8	forward	5′-CCAGGCTGAGAAGGGTGATA-3′
	reverse	5′-GCAGGAAGAGTGTCTGTGGC-3′
hIes6	forward	5′-ATGGCGGCGCAAATTCCAAT-3′
	reverse	5′-AATGGCAAAGGTTTGGCAGC-3′
hIes2	forward	5′-GGAGAAGCCCTGGAGTTGAG-3′
	reverse	5′-GGAACACTCTTGGTCCCCAG-3′
GAPDH	forward	5′-ATCACTGCCACCCAGAAGAC-3′
	reverse	5′-ATGAGGTCCACCACCCTGTT-3′
YY1	forward	5′-CCCTCATAAAGGCTGCACAA-3′
	reverse	5′-TGAACCAGTTGGTGTCGTTT-3′
hArp5 (BC038402)	forward	5′-CTACATCCAGAAGCTCAGTAT-3′
	reverse	5′-CTTCTCCTATGAGAGATGGCT-3′
miR-372-3p	forward	5′-TAGCAGGATGGCCCTAGACC-3′
	reverse	5′-TCCGTTGATATGGGCGTCAC-3′
miR-372-3p-RT	forward	5′-ACACTCCAGCTGGGAAAGTGCTGCGACATTT-3′
	reverse	5′-GTGCAGGGTCCGAGGT-3′

## Data Availability

The microarray datasets are made available through the National Center for Biotechnology Information’s Gene Expression Omnibus (GEO) data repository (http://www.ncbi.nlm.nih.gov/geo/) using the series accession number GSE68655.

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
