# Peer review of "Feedback Modulation between Human INO80 Chromatin Remodeling Complex and miR-372 in HCT116 Cells"

_ijms, 2023, doi:10.3390/ijms241310685_

Round 1
Reviewer 1 Report
This paper describes a mutual regulatory mechanism between miR-372 and the human INO80 chromatin remodeling complex (INO80 complex). The authors have proved this mutual regulation by various knock-down and overexpression experiments in HeLa cell and HCT116 colorectal cancer cell lines. INO80 complex was shown to be involved in basic biological functions like cell proliferation, cell adhesion, cell growth, cell migration, etc. miR-372 is also shown to affect the p21/p53 pathway, which is downstream of the INO80 complex. Though there have been several reports showing the role of miR-372 in various biological processes. The authors here are showing the regulation of an important chromatin remodeling complex, INO80 complex, which can affect several cellular housekeeping functions in an organism. However, the authors need to address the following points before it can be accepted for publication.
1. The introduction of the paper can be improved. The authors have mentioned the role of miR-372 throughout the introduction in various paragraphs. However, they can use colorectal cancer and microRNA regulation part first and move all miR-372-related studies into one paragraph before moving on to the objective of this paper.
2. Line 49-50: The authors should cite a review addressing the role of miRNA in important biological processes instead of a specific paper (ref 7) on miR-17-3p contributing to exercise-induced cardiac growth.
3. In Figure 2, Panel A: Dox dosages of 1ug and 2ug (shown in lanes 3 and 4) do not look different. 1ug dose seems to have lower INO80 than 2ug Dox.
4. In Figure 2, Panel B: The upper and lower panels should be marked. I guess the lower panels are wild-type as per figure legends. Does wild type mean that cells were not induced by Dox? The figure should be marked appropriately.
5. In Figure 2, Panel C: X-axis is missing in the lower graph.
6. In Figure 2, Panel D showed the knock-down of INO80 inhibits cell proliferation. In contrast, the knock-down of INO80 by Dox in HeLa cells causes cells to overgrow in clusters (Panel B). Can the authors discuss this discrepancy, whether it is the effects of Dox or is this effect cell line specific?
7. Figure 3: miR-372/373 were detected in knock-downs of INO80, Arp8, Arp5, les2, and les6 in HeLa cells. On what basis, only Arp8 and les2 were selected for confirmation of knockdown and miR-372 levels in HCT116 cells? Similarly, on what basis subunits YY1 and INO80 were selected and others were left?
8. Figure 4, Panel A: why different concentrations of vectors were used for checking the efficiency of miR-372? Panel C: why the BCCIP antibody was used?
9. miR-372 sites were found in 3’UTRs of INO80, YY1, ARP8 and MCRS1, but why did only INO80 and Arp8 were chosen for luciferase assays?
10. Figure 5: Panel A: why seed sites of miR-372 were not mutated for Arp8 3’UTR?
11. Figure 5: Panel B: what do Arp8 3’UTR-1, Arp8 3’UTR-2, INO80 3’UTR-1 and INO80 3’UTR-2 indicate? Does that mean that seed site 1 is mutated in Arp8 3’UTR-1 or only part of 3’UTR containing seed 1 was used? Figure legends, text and material and methods do not suggest that. If they are using just wild type 3’UTR, then Arp8 3’UTR-1 and Arp8 3’UTR-2 should be the same. Why is there so much difference in luciferase activity if wild type 3’UTR was used? Why was mutated 3’UTR not shown in the luciferase assay in panel B. Mutated version of INO80 3’UTR is shown in panel E in a very confusing way. Lanes 1 and 2 are empty vector controls, but why do they have INO80 bands? Does that suggest endogenous INO80 protein levels? Mutated 3’UTR should be shown along with empty vectors control and wild type 3’UTR so that a proper comparison can be made.
12. Figure 6, Panel B: X-axis is missing from the lower panel graphs.
Overall English is fine except some grammar mistakes were observed in lines 19, 183, 400, 411, 417, 496, etc.
Author Response
The authors are thankful to the reviewer for understanding the purpose of this manuscript, making comments, and giving valuable suggestions to improve the quality of the manuscript.
Review 1:
The authors are thankful to the reviewer for understanding the purpose of this manuscript, making comments, and giving valuable suggestions to improve the quality of the manuscript.
Comments and Suggestions for Authors
This paper describes a mutual regulatory mechanism between miR-372 and the human INO80 chromatin remodeling complex (INO80 complex). The authors have proved this mutual regulation by various knock-down and overexpression experiments in HeLa cell and HCT116 colorectal cancer cell lines. INO80 complex was shown to be involved in basic biological functions like cell proliferation, cell adhesion, cell growth, cell migration, etc. miR-372 is also shown to affect the p21/p53 pathway, which is downstream of the INO80 complex. Though there have been several reports showing the role of miR-372 in various biological processes. The authors here are showing the regulation of an important chromatin remodeling complex, INO80 complex, which can affect several cellular housekeeping functions in an organism. However, the authors need to address the following points before it can be accepted for publication.
- The introduction of the paper can be improved. The authors have mentioned the role of miR-372 throughout the introduction in various paragraphs. However, they can use colorectal cancer and microRNA regulation part first and move all miR-372-related studies into one paragraph before moving on to the objective of this paper.
We agree with the reviewer's suggestion. Based on the reviewer’s comments, we rearranged the contents in the introduction to enhance its clarity and flow. In revised MS, we prioritized discussing colorectal cancer and microRNA regulation at the beginning of the introduction. Additionally, we consolidated all miR-372-related studies into one paragraph, providing a concise overview of the relevant literature before moving on to the objective of our paper.
- Line 49-50: The authors should cite a review addressing the role of miRNA in important biological processes instead of a specific paper (ref 7) on miR-17-3p contributing to exercise-induced cardiac growth.
We fully agree with the reviewer's suggestion. We have replaced the reference 7 with an appropriate review article that summarizes the broader understanding of miRNA involvement in essential biological processes.
- In Figure 2, Panel A: Dox dosages of 1ug and 2ug (shown in lanes 3 and 4) do not look different. 1ug dose seems to have lower INO80 than 2ug Dox.
Thank you for the reviewer's reminder. To address this concern, we quantified and statistically analyzed INO80 protein levels by normalizing to the corresponding internal protein GAPDH using ImageJ software. In revised MS, quantified data added to Figure 2A (lower panel).
- In Figure 2, Panel B: The upper and lower panels should be marked. I guess the lower panels are wild-type as per figure legends. Does wild type mean that cells were not induced by Dox? The figure should be marked appropriately.
We appreciate the reviewer's feedback regarding Figure 2B. Lower panel indicates the pSingle-shNT cells, and cells were subjected to Dox induction under the same conditions. In revised MS, we have marked the lower panel images appropriately.
- In Figure 2, Panel C: X-axis is missing in the lower graph.
Thank you for the reviewer's reminder. In revised MS, the X-axis added in Figure 2C lower graph.
- In Figure 2, Panel D showed the knock-down of INO80 inhibits cell proliferation. In contrast, the knock-down of INO80 by Dox in HeLa cells causes cells to overgrow in clusters (Panel B). Can the authors discuss this discrepancy, whether it is the effects of Dox or is this effect cell line specific?
We appreciate the questions raised by the reviewer. Firstly, we believe that cell growth and clone formation ability are two different cellular behaviors with different mechanisms involved. Under our experimental conditions, knockdown of INO80 induced by Dox resulted in aggregated growth of cells, and the cells overlying it were prone to death. And the ability to form clones is more representative of the cell's ability to proliferation.
- Figure 3: miR-372/373 were detected in knock-downs of INO80, Arp8, Arp5, les2, and les6 in HeLa cells. On what basis, only Arp8 and les2 were selected for confirmation of knockdown and miR-372 levels in HCT116 cells? Similarly, on what basis subunits YY1 and INO80 were selected and others were left?
We understand the questions raised by the reviewer. INO80 chromatin remodeling complex is a multiprotein enzyme containing 15 subunits. As shown in Figure 1A, HSA and SNF2 modules play a key role in maintaining the holoenzyme activities. The absence of either subunit in these two modules may affect the DNA sliding activity and the ATPase activity of the INO80 complex. Therefore, we selected representative key subunits from different modules for knockdown and performed DAN microarray sequencing. By screening candidate target genes that are jointly regulated by these subunits, such as miR-372/373, as the target regulatory genes for the entire INO80 complex.
In subsequent experiments, we focused more on factors that affect the activities of the overall complex, such as Arp8 and Ies2, which come from different modules. Of course, INO80 protein is a catalytic subunit, and changes in its expression level can directly affect the enzymatic activity of INO80 complex. We chose YY1 because our Lab has already constructed overexpression and knockdown plasmids in other studies.
- Figure 4, Panel A: why different concentrations of vectors were used for checking the efficiency of miR-372? Panel C: why the BCCIP antibody was used?
We appreciate the questions raised by the reviewer. The use of different concentrations of vectors is to ensure consistency in the total amount of plasmids contained in each dish of cell culture, thereby avoiding the potential impact on cell growth due to the different total amount of plasmids added to each dish of cells.
In Figure C, BCCIP only wants to demonstrate that miR-372 does not have inhibitory effects on all genes, but selectively targets certain genes.
- miR-372 sites were found in 3’UTRs of INO80, YY1, ARP8 and MCRS1, but why did only INO80 and Arp8 were chosen for luciferase assays?
We understand the questions raised by the reviewer. As mentioned in question 7, the INO80 complex is a complex containing 15 subunits, and miR-372 inhibits the catalytic subunit INO80 or Arp8 subunit in the HSA module, which can affect the overall function of the INO80 complex. Therefore, we speculate that miR-372 acts on the 3 '- UTR region of other subunits, with a similar effect.
- Figure 5: Panel A: why seed sites of miR-372 were not mutated for Arp8 3’UTR?
Thank you to the reviewer for the question. Of course, mutation of the seed sites of miR-372 in Arp8-3’UTR is more persuasive. Actually, we have designed and attempted to construct the plasmid, but there were problems during the plasmid construction process and the mutant plasmid was not obtained in a timely manner.
- Figure 5: Panel B: what do Arp8 3’UTR-1, Arp8 3’UTR-2, INO80 3’UTR-1 and INO80 3’UTR-2 indicate? Does that mean that seed site 1 is mutated in Arp8 3’UTR-1 or only part of 3’UTR containing seed 1 was used? Figure legends, text and material and methods do not suggest that. If they are using just wild type 3’UTR, then Arp8 3’UTR-1 and Arp8 3’UTR-2 should be the same. Why is there so much difference in luciferase activity if wild type 3’UTR was used? Why was mutated 3’UTR not shown in the luciferase assay in panel B.
We appreciate the questions raised by the reviewer. In Figure 5B: Arp8 3'UTR-1, Arp8 3'UTR-2, INO80 3'UTR-1, and INO80 3'UTR-2 represent two miR-372 seed sites in the 3'UTR region of Arp8 and INO80, one near the C-terminus of the mRNA and the other far from the C-terminus of the mRNA. Perhaps this led to the difference in Luciferase activity. In the Luciferase assay of group B, only two wild type activities were detected.
Mutated version of INO80 3’UTR is shown in panel E in a very confusing way. Lanes 1 and 2 are empty vector controls, but why do they have INO80 bands? Does that suggest endogenous INO80 protein levels? Mutated 3’UTR should be shown along with empty vectors control and wild type 3’UTR so that a proper comparison can be made.
We appreciate the questions raised by the reviewer. We are very sorry for the confusion caused by our unclear marks. Lanes 1 and 2 are empty vector control groups, and the protein bands are the endogenous INO80 protein levels. We fully agree to the reviewer's statement. The best experimental design is to display the mutant 3'UTR together with the empty control and wild-type 3'UTR. Although we conducted separate experiments, miR-372 significantly inhibited the INO80 protein levels caused by transfection of WT in C and D.
- Figure 6, Panel B: X-axis is missing from the lower panel graphs.
Thank you for the reviewer's reminder. In revised MS, the X-axis added in Figure 6B lower panel graph.
Comments on the Quality of English Language
Overall English is fine except some grammar mistakes were observed in lines 19, 183, 400, 411, 417, 496, etc.
Thank you for the reviewer's reminder. We have made corrections in the grammar mistakes observed in lines 19, 183, 400, 411, 417, 496, and other sections of the paper.
The proper English language has been extensively revised by English-speaker.

Reviewer 2 Report
Peer-reviewed paper by Shah et al. concerns a proposed mutual regulatory interplay between INO80 multiprotein chromatin remodeling complex and miR-372 microRNA. In the manuscript, authors provide comprehensive evidence of this mutual regulation by using a broad spectrum of the highly relevant molecular tools, such as siRNA and shRNA knockdown of INO80 components, miR-372 overexpresson, luciferase reporter system, etc. While the above-mentioned mechanistic data to support the idea of the mutual regulation between INO80 and miR-372 are convincing, their functional interpretation in context of the cancer rises some questions. I have a few comments to be addressed, before the paper can be published.
1. In the introductory section, authors briefly consider the role of microRNAs, including miR-372, in the cancerogenesis. With that, they say nothing at all about the INO80 there. It remains absolutely unclear to the reader, why the authors decided further to look for a connection between the two molecules. I would suggest to clarify this and transfer some INO80-related literature data from the Discussion to Introduction.
2. Part 2.1. “DNA Microarray Analysis…” is obviously uninformative. To be honest, Fig.1c and 1d say nearly nothing to the biologist. To my vision, it would be helpful to specify which genes are up- or down- regulated, perhaps as a supplementary data set (which is quite routine for this kind of transcriptomics research). Also, the gene ontology (Fig 1d) looks very poor without further detalization. Finally, what is, for instance, the difference between the “cell migration” and “cell motion” on Fig 1d? Or between the “cell proliferation” and “cell division”? – I can not understand this. This section should be either seriously revised and expanded, or just excluded from the paper.
3. Part 2.2., Fig 2.B. This is unclear to me. On the upper image “before induction” is shown normal, nearly confluent cell monolayer. Then, there are images of 1 and 2 weeks of Dox-induction, where one can see monolayer destruction and separate groups of cells. Authors interpret this as an “abnormal cell growth and proliferation”. Why? As I understand, authors did not trypsinize and passage cells within these 2 weeks. In the confluent monolayer, cells have no option to divide. So, based on these data it is in principle hard to conclude anything about the proliferation. Looking at them, I would rather speak about the partial cell death (perhaps via apoptosis) as a result of INO80 knockdown there. By the way, the same is related to the colony formation results, too.
There are many specific assays both for the proliferation (in terms of oncology, ki-67 index is highly relevant), and for the cell viability and apoptosis. These aspects are of high significance for the molecules under study, and these new results definitely would strengthen the paper. I would suggest to perform this job.
4. In relation to above. When authors consider p53/p21 pathway, they also totally ignore its role in the apoptosis. Meanwhile, there are many data that these proteins are involved into apoptotic pathways, I would recommend to discuss this point.
5. English is not perfect, some phrases are built in the hardly understandable way. For example, in the abstract: «Herein, we showed evidence that the INO80 complex transcriptionally controls microRNA-372 (miR-372) expression through RNA-Seq and biological experiments». One can think that INO80 controls miR-372 through RNA-Seq.
Or: “MiR-372 may operate as tumor suppressors by inhibiting oncogene expression or as oncogenic miRNAs by suppressing the expression of tumor suppressor genes". Too much expression and suppression in one sentence. Better to re-write.
English needs an improvement by the professional proofreading.
Author Response
The authors are thankful to the reviewer for understanding the purpose of this manuscript, making comments, and giving valuable suggestions to improve the quality of the manuscript.
Review-2:
Comments and Suggestions for Authors
Peer-reviewed paper by Shah et al. concerns a proposed mutual regulatory interplay between INO80 multiprotein chromatin remodeling complex and miR-372 microRNA. In the manuscript, authors provide comprehensive evidence of this mutual regulation by using a broad spectrum of the highly relevant molecular tools, such as siRNA and shRNA knockdown of INO80 components, miR-372 overexpresson, luciferase reporter system, etc. While the above-mentioned mechanistic data to support the idea of the mutual regulation between INO80 and miR-372 are convincing, their functional interpretation in context of the cancer rises some questions. I have a few comments to be addressed, before the paper can be published.
- In the introductory section, authors briefly consider the role of microRNAs, including miR-372, in the cancerogenesis. With that, they say nothing at all about the INO80 there. It remains absolutely unclear to the reader, why the authors decided further to look for a connectionbetween the two molecules. I would suggest to clarify this and transfer some INO80-related literature data from the Discussion to Introduction.
We appreciate your feedback and understand your concerns about the clear connection between miR-372 and INO80 in the context of carcinogenesis. We have carefully revised the introduction section to provide a clearer explanation between miR-372 and INO80. In addition, we also provided relevant citations on INO80.
- Part 2.1. “DNA Microarray Analysis…” is obviously uninformative. To be honest, Fig.1c and 1d say nearly nothing to the biologist. To my vision, it would be helpful to specify which genes are up- or down- regulated, perhaps as a supplementary data set (which is quite routine for this kind of transcriptomics research). Also, the gene ontology (Fig 1d) looks very poor without further detalization. Finally, what is, for instance, the difference between the “cell migration” and “cell motion” on Fig 1d? Or between the “cell proliferation” and “cell division”? – I can not understand this. This section should be either seriously revised and expanded, or just excluded from the paper.
Thank you for the key questions raised by the reviewers and agree with their suggestions. In order to provide more information, we have added access information for the microarray data, so that the people interested in this data can directly obtain the relevant information in the database. We have added following sentences in 2.1 section: “The Illumina microarray datasets are accessible from the National Center for Biotechnology Information Gene Expression Omnibus (GEO) data repository (accession#: GSE68655)”.
- Part 2.2., Fig 2.B. This is unclear to me. On the upper image “before induction” is shown normal, nearly confluent cell monolayer. Then, there are images of 1 and 2 weeks of Dox-induction, where one can see monolayer destruction and separate groups of cells. Authors interpret this as an “abnormal cell growth and proliferation”. Why? As I understand, authors did not trypsinize and passage cells within these 2 weeks. In the confluent monolayer, cells have no option to divide. So, based on these data it is in principle hard to conclude anything about the proliferation. Looking at them, I would rather speak about the partial cell death (perhaps via apoptosis) as a result of INO80 knockdown there. By the way, the same is related to the colony formation results, too.
There are many specific assays both for the proliferation (in terms of oncology, ki-67 index is highly relevant), and for the cell viability and apoptosis. These aspects are of high significance for the molecules under study, and these new results definitely would strengthen the paper. I would suggest to perform this job.
We understand the questions raised by the reviewer. From the image, it seems to be a simple cell monolayer destruction and separation of cell populations. However, under the microscope, it can be clearly seen that Dox induced INO80 knockdown leads to morphological changes in cells, especially leading to failure of cytoplasmic division. Fluorescence staining of the cells can also reveal many multipolar spindle phenomena. Therefore, we described in the text that "..., suggesting the involvement of the INO80 complex in regulating the normal cell growth process ". In fact, we can also see from the images that there are differences in the growth status of cells before induction, the first week, and the second week. Of course, as reviewer have pointed out, by the second week, many cells may die through apoptosis, which may also be related to a decrease in colony formation.
We are also very grateful to the reviewers for their suggestions on detecting and estimating cell proliferation related indicators and experiments. We will add these experimental contents in the future in-depth study of molecular mechanisms.
- In relation to above. When authors consider p53/p21 pathway, they also totally ignore its role in the apoptosis. Meanwhile, there are many data that these proteins are involved into apoptotic pathways, I would recommend to discuss this point.
Thank you for your comment regarding the role of the p53/p21 pathway in apoptosis. We appreciate your insight and suggestion to discuss this important point in our paper. To address this concern, we have incorporated the appropriate information and references to highlight the connection between the p53/p21 pathway and apoptosis. (See beginning of the 2.6 section).
- English is not perfect, some phrases are built in the hardly understandable way. For example, in the abstract: «Herein, we showed evidence that the INO80 complex transcriptionally controls microRNA-372 (miR-372) expression through RNA-Seq and biological experiments». One can think that INO80 controls miR-372 through RNA-Seq.
Or: “MiR-372 may operate as tumor suppressors by inhibiting oncogene expression or as oncogenic miRNAs by suppressing the expression of tumor suppressor genes". Too much expression and suppression in one sentence. Better to re-write.
Thank you for your valuable comments. We have carefully considered your suggestions and made appropriate changes to the sentences accordingly."
English needs an improvement by the professional proofreading.
The proper English language has been extensively revised by a native English-speaker.

Reviewer 3 Report
Ali Shah et al describes the interrelationship between INO80 chromatin remodeling complex and micro-RNA miR-372.
They performed all the right experiments for this kind of research, including knocking down subunits of INO80 (INO80, Arp5, Arp8, Ies2 and Ies6), microarray analysis, GO ontology, stably transfected induce-silencing sh-plasmids, colony formation, FACS, cell cycle, incubation with pri-miR, RNA and protein levels, and mutagenesis of 3'-UTR sites target of miRNAs.
They conclude that the INO80 complex and miR-372 are mutually regulated in HCT116 colon cancer cells.
Comments:
1) It is not clear what kind of microarray analysis was performed. DNA in general ?, specific for miRNAs, or both ?.
2) Results from the microarrays should be uploaded to a public database, such as GEO and give an series accesion number.
3) Are there cellular genes in common among the differentially expressed genes upon silencing of the different subunits, or only the miRNAs in Figure 3A. It would be useful to include a Venn diagram with all the intersections between the differentially expressed genes after silencing each subunit of INO80 (the same way the authors did for figure 1C of reference #34 of the present manuscript.
4) miR-611 also looks promising. Did the authors performe experimentation with this miRNA ?
5) I could not find in reference #34 your statement written in line 335-336 that "INO80 complex upregulated miR-372 in HeLa cells". I even performed using the software Geo2R the analysis of the 250 most differentially expressed genes in accesion # GSE68655 (in the GEO database) not finding miR-372. Could you verify this statement and explain more clearly the source of the data.
6) It is clear from the results that miR-372 post-transcriptionally regulates the expression of the INO80 complex.
However, It is not very clear the mechanism by which knocking down INO80 produces an increase in miR-372.
I believe that the authors could write in a more clear way their proposed mechanism as to how the downregulation of INO80 by the different siRNAs can increase the expression of miR-372, which in turn could downregulate the expression of INO80.
7) Figure 2D: It is very surprising the low percentage of cells in G1 in the control cells (vector alone) and also the high proportion in S phase, compared with publication using HCT116 colon cancer cells. Please explain.
Minor:
line 46. Introduction of miRNAs would read better as: "MicroRNAs (miRNA) are non-coding single stranded RNAs with an average of 22 nucleotides present in plants, animals and viruses"... presented in plural as a class of RNAs, not as a single one.
Lines 97-104 in "Results" actually belong to the Introduction section. Results should start with the sentence: "To investigate the target genes...(in line 104)".
Line 110: that "respectively" does not mean anything
Minor modifications reviewed by a native English
Author Response
The authors are thankful to the reviewer for understanding the purpose of this manuscript, making comments, and giving valuable suggestions to improve the quality of the manuscript.
Comments and Suggestions for Authors
Ali Shah et al describes the interrelationship between INO80 chromatin remodeling complex and micro-RNA miR-372. They performed all the right experiments for this kind of research, including knocking down subunits of INO80 (INO80, Arp5, Arp8, Ies2 and Ies6), microarray analysis, GO ontology, stably transfected induce-silencing sh-plasmids, colony formation, FACS, cell cycle, incubation with pri-miR, RNA and protein levels, and mutagenesis of 3'-UTR sites target of miRNAs. They conclude that the INO80 complex and miR-372 are mutually regulated in HCT116 colon cancer cells.
Comments:
1) It is not clear what kind of microarray analysis was performed. DNA in general ?, specific for miRNAs, or both ?.
The DNA microarray analysis was conducted with DNA in general.
2) Results from the microarrays should be uploaded to a public database, such as GEO and give an series accesion number.
Thank you for the reviewer's reminder. We added the sentences “The Illumina microarray datasets are accessible from the National Center for Biotechnology Information Gene Expression Omnibus (GEO) data repository (http://www.ncbi.nlm.nih.gov/geo/) using the series accession number GSE68655.
3) Are there cellular genes in common among the differentially expressed genes upon silencing of the different subunits, or only the miRNAs in Figure 3A. It would be useful to include a Venn diagram with all the intersections between the differentially expressed genes after silencing each subunit of INO80 (the same way the authors did for figure 1C of reference #34 of the present manuscript.
Of course, there are common cellular genes among differentially expressed genes silenced by different subunits, and only some co-regulated miRNAs are shown here. Due to the focus of this study on the correlation between miR-372 and INO80 complex, other overlapping genes were not listed.
4) miR-611 also looks promising. Did the authors performe experimentation with this miRNA ?
Thank you for your comment and interest in our study. Although miR-611 appears in the gene expression profiles of knocking down INO80, Arp8, and Ies6 subunits, we also predict that miR-611 may also be regulated by the INO80 complex. However, we have not yet truly investigated the interaction mechanism between miR-611 and INO80 complexes.
5) I could not find in reference #34 your statement written in line 335-336 that "INO80 complex upregulated miR-372 in HeLa cells". I even performed using the software Geo2R the analysis of the 250 most differentially expressed genes in accesion # GSE68655 (in the GEO database) not finding miR-372. Could you verify this statement and explain more clearly the source of the data.
Thank you for the reviewer's reminder. This is our mistake, In the end of sentence, we removed [34], and added (http://www.ncbi.nlm.nih.gov/geo/, accession number GSE68655).
6) It is clear from the results that miR-372 post-transcriptionally regulates the expression of the INO80 complex. However, It is not very clear the mechanism by which knocking down INO80 produces an increase in miR-372.
I believe that the authors could write in a more clear way their proposed mechanism as to how the downregulation of INO80 by the different siRNAs can increase the expression of miR-372, which in turn could downregulate the expression of INO80.
We understand the questions raised by the reviewer. However, although we do not have direct evidence to explain the regulation of miR-372 expression by the INO80 complex; but the regulatory effect of INO80 on miR-372 can be inferred from the following two aspects: 1) miR-372 was found in the gene expression profiles after knocking down multiple subunits of the INO80 complex, suggesting that miR-372 may be a targeted regulatory gene for the INO80 complex; 2) It has been reported that YY1, one of the subunit of the INO80 complex, participates in making up the HSA module, can recruit and bind to the promoter region of miR-372. Therefore, we speculate that YY1, as a transcription factor, may bring the INO80 complex to the miR-372 promoter region, resulting in knocking down multiple subunits that can increase the expression level of miR-372.
7) Figure 2D: It is very surprising the low percentage of cells in G1 in the control cells (vector alone) and also the high proportion in S phase, compared with publication using HCT116 colon cancer cells. Please explain.
We understand the questions raised by the reviewer. However, under our experimental conditions, we conducted three experiments with a similar tendency. It cannot be ruled out that it is related to the pLVX vector, but a more precise reason cannot be explained without the basis of experimental data.
Minor:
line 46. Introduction of miRNAs would read better as: "MicroRNAs (miRNA) are non-coding single stranded RNAs with an average of 22 nucleotides present in plants, animals and viruses"... presented in plural as a class of RNAs, not as a single one.
Thank you for your comment and suggestion regarding the introduction of miRNAs. We appreciate your feedback, and we agree that it is more appropriate to present miRNAs as a class of RNAs rather than a single entity. We have made the necessary revision to the sentence as follows: "MicroRNAs (miRNAs) are non-coding single-stranded RNAs".
Lines 97-104 in "Results" actually belong to the Introduction section. Results should start with the sentence: "To investigate the target genes...(in line 104)".
We agree with the reviewer suggestion. Following the reviewer's suggestion, we have moved lines 97-104 to the introduction section.
Line 110: that "respectively" does not mean anything
Thank you for your feedback regarding the use of the term "respectively" in line 110. We have removed “respectively” from line 110.
Comments on the Quality of English Language
Minor modifications reviewed by a native English-speaker.

Reviewer 4 Report
Present article by Shah et al. titled "Feedback modulation between human INO80 chromatin re-2 modeling complex and miR-372 in HCT116 cells" is scientifically sound and well presented. I have following comments:
1. Authors have used two different cell lines in the present study "HCT116 colon cancer and HeLa cervical cancer cell lines" to study human INO80 chromatin remodeling complex and miR-372. Both cell lines have its own genetic architecture that can influence the expression level of various genes. Please explain the cell line selection criteria.
2. The target relationship between miR-372 and INO80 complex was verified in HCT116 colon cancer cells only but the He la cells did not confirm this relationship. Does it have to do something with the difference in etiopathogenesis of each cancer?
3. Line 127 The authors have stated that "The Silencing of INO80 Caused HeLa Cells to Grow in Clusters and Inhibited the Colony Forming Ability of HCT116 Cells" means that INO80 silencing has nearly opposite effect on two different cell lines (Hela cells are growing more and HKT116 cells growth is restricted). What can be the possible explanation behind this?
Author Response
The authors are thankful to the reviewer for understanding the purpose of this manuscript, making comments, and giving valuable suggestions to improve the quality of the manuscript.
Comments and Suggestions for Authors
Present article by Shah et al. titled "Feedback modulation between human INO80 chromatin re-2 modeling complex and miR-372 in HCT116 cells" is scientifically sound and well presented. I have following comments:
- Authors have used two different cell lines in the present study "HCT116 colon cancer and HeLa cervical cancer cell lines" to study human INO80 chromatin remodeling complex and miR-372. Both cell lines have its own genetic architecture that can influence the expression level of various genes. Please explain the cell line selection criteria.
The authors are thankful to the reviewer for understanding the purpose of this manuscript, making comments, and giving valuable suggestions to improve the quality of the manuscript.
We agree with the reviewer's opinion. Actually, DNA microarrays was conducted as part of a previous research project. Later, statistical analysis revealed that miR-372/373 may be regulated by the INO80 complex. Subsequently, we detected the expression of miR-372 in multiple different tumor tissues and found that the expression of miR-372 was significantly reduced in colon cancer. Therefore, in this research we used HCT116 cells.
- The target relationship between miR-372 and INO80 complex was verified in HCT116 colon cancer cells only but the Hela cells did not confirm this relationship. Does it have to do something with the difference in etiopathogenesis of each cancer?
Thank you for the reviewer's question. In fact, when we used clinical tumor samples to detect the expression level of miR-372, we found that there was no significant difference in the expression level of miR-372 in cervical cancer compared to the control group. However, a significant decrease was found in colon cancer. Therefore, we agree with the reviewer's explanation that the correlation between miR-372 and INO80 may vary in different tissues or cell lines due to the different causes of each cancer
- Line 127 The authors have stated that "The Silencing of INO80 Caused HeLa Cells to Grow in Clusters and Inhibited the Colony Forming Ability of HCT116 Cells" means that INO80 silencing has nearly opposite effect on two different cell lines (Hela cells are growing more and HKT116 cells growth is restricted). What can be the possible explanation behind this?
We appreciate the questions raised by the reviewer. Firstly, we believe that cell growth and clone formation ability are two different cellular behaviors with different mechanisms involved. Under our experimental conditions, knockdown of INO80 induced by Dox resulted in aggregated growth of cells, and the cells overlying it were prone to death. And the ability to form clones is more representative of the cell's ability to proliferate.

Round 2
Reviewer 2 Report
The authors payed attention to all of my comments, the paper has been improved substantially. It is ready for publishing now.
Author Response
Thank you to the reviewer for affirming our response content and manuscript.
Reviewer 3 Report
Most of the answers by the authors are fine except for the analysis of the microarray. Now, they state that the GEO accesion numer is GSE68655. Accordingly, I accessed this dataset and proceeded to perform the analysis using the Geo2R software within the GEO homepage itself.
Following the recommended procedure by GEO I calculated the logFC, that is (expression value of the tests divided by the control).
The result is that the levels of expression of miR373, 611, 1206, 1204 and 614 are downregulated, and not overexpressed.
I am including a table with also the values of Fold Change
LogFC | FC | Gene Symbol |
-3.7309901 | 0.07531129 | MIR373 |
-1.00317809 | 0.49889977 | MIR611 |
-0.37137022 | 0.77304793 | MIR1206 |
-0.65109064 | 0.63679873 | MIR1204 |
-0.29555821 | 0.81475702 | MIR614 |
Whereas the expression values of INO80 subunits are underexpressed, meaning that the knocking down by the interference RNA experiments was OK, but the expression of miRNAs was underregulated, instead of upregulated.
Also miR372 did not appear in the list.
I suggest that the authors revise this point, and check with the values they uploaded to GEO.
English is ok with some minor corrections needed by a native English speaker
Author Response
The authors are thankful to the reviewer for understanding the purpose of this manuscript, making comments, and giving valuable suggestions to improve the quality of the manuscript.
Comments and Suggestions for Authors
Most of the answers by the authors are fine except for the analysis of the microarray. Now, they state that the GEO accesion numer is GSE68655. Accordingly, I accessed this dataset and proceeded to perform the analysis using the Geo2R software within the GEO homepage itself.
Following the recommended procedure by GEO I calculated the logFC, that is (expression value of the tests divided by the control).
The result is that the levels of expression of miR373, 611, 1206, 1204 and 614 are downregulated, and not overexpressed.
I am including a table with also the values of Fold Change
LogFC |
FC |
Gene Symbol |
-3.7309901 |
0.07531129 |
MIR373 |
-1.00317809 |
0.49889977 |
MIR611 |
-0.37137022 |
0.77304793 |
MIR1206 |
-0.65109064 |
0.63679873 |
MIR1204 |
-0.29555821 |
0.81475702 |
MIR614 |
Whereas the expression values of INO80 subunits are underexpressed, meaning that the knocking down by the interference RNA experiments was OK, but the expression of miRNAs was underregulated, instead of upregulated.
Also miR372 did not appear in the list.
I suggest that the authors revise this point, and check with the values they uploaded to GEO.
- We appreciate the questions raised by the reviewer. The Microarray experiments and analysis were performed: High-quality RNA samples were converted to cDNA and biotin-labeled for microarray analysis using Ambion’s Illumina TotalPrep RNA Amplification Kit (Life Technologies, Grand Island, NY, USA). Labeled cRNAs were processed on a HumanHT-12 v4 Expression BeadChip (Illumina, San Diego, CA, USA) and imaged using an Illumina iScan system by the company that conducted the microarray experiments. This chip targets 448 000 probes that provide genome-wide coverage of well-characterized genes, gene candidates, and splice variants. We implemented three steps to identify the probes that the Ino80 complex may influence. Firstly, Quartile normalization and log transform were performed before the analysis. After that, to find differentially expressed genes (DEGs) between the knock-down samples (for subunits of hIno80) and siNT control, the Illumina Custom differential expression algorithm was used to test the mean difference and to evaluate differentially expressed genes. Then, the probes influenced by the INO80 complex were selected based on whether the value of one probe was different in any hINO80 knockdown subunits (at least in one subunit) with the corresponding value in no-targeting cells. MIR372 and MIR373 were down-regulated in our microarray data (Please refer to the table below). Thus we described these miRNAs as “identified” In figure 3A. However, this phenotype was different in HCT116 cells than in HeLa cells. In addition, it is worth noting that the average signal of MIR372 expression in INO80-knockdown HeLa cells was a minus value (-0.61). Thus, it would not be screened out by LogFC since the Log of a minus value is meaningless in this context. That’s why miR372 did not appear in the Geo2R list of DEGs.
The possible reason is that the INO80 complex, as a chromatin remodeling complex, is widely involved in the transcriptional regulation of multiple genes by altering chromatin structure. Upon knocking down the INO80 complex in different types of cancer cells, the proteins (or complexes) or transcription cofactors recruited by INO80 in the promoter region of the miR-371-373 gene cluster may have different effects on the expression level of miR-372-373; Furthermore, according to the currently reported literature, the expression level of miR-372 varies depending on the type of cancer. For example, compared to normal tissues, miR-372-373 is significantly reduced in tissues such as non-small cell lung cancer, pancreatic cancer, cervical cancer, endometrial cancer, and ovarian cancer. On the contrary, the expression of miR-372 is higher in lung squamous cell carcinoma, colorectal cancer, esophageal squamous cell carcinoma, and hepatocellular carcinoma than in normal tissues. It is worth mentioning that in certain specific cancer types, the expression level of miR-372 contradicts the results reported by different research groups, such as breast cancer and colon cancer. Under our experimental conditions, we found that miR-372 expression was significantly reduced in colon cancer tissue compared to normal tissue. The expression status of miRNAs in different cancer cells may be closely related to the target genes they interact with, highlighting the complexity of the molecular mechanisms of miRNAs in tumor development in different cancer.
Comments on the Quality of English Language
English is ok, with some minor corrections needed by a native English speaker.
- Native English-speaker has extensively revised the proper English language.

Round 3
Reviewer 3 Report
I checked the new information supplied by the authors. I found that the description they give about the characteristics of the microarray used ("This chip targets 448 000 probes that provide genome-wide coverage of well-characterized genes, gene candidates, and splice variants") does not correspond with the information that appears in the GEO database of the NCBI.
In fact the real number is: Each array on the HumanHT-12 v4 Expression BeadChip targets more than 31,000 annotated genes with more than 47,000 probes derived from the National Center for Biotechnology Information Reference Sequence (NCBI), which is the information that can be found also in the Illumina webpage.
- This microarray accession number correspond to HeLa cells, and according to my previous comments the authors recognize that miR373 was downregulated (also miR372).
However, it turns out that in HCT116 cells, the effect on these miRNAs is just the opposite, as they observe after knocking down members of the INO80 complex (Ies2 and Arp)with siRNAs, and posteriorly with siRNA against YY1 and INO80 with an shRNA. In all these cases pri-miR-372 was upregulated, as they show in Figure 3c though 3i.
However, in Figure 3A labeled as "Identified miRNAs in knockdown gene expression profiles" there is no distinction among the miRNAs down-or up-regulated. They appear as "identified". That figure 3A correspond to the microarray performed in HeLa cells, and the rest of the graphs of Figure 3 correspond to HCT116 cells.
I think that the present way of expressing these results is confusing. It should be clearly stated what the effects on these miRNAs in Fig 3A correspond HeLa cells and then clearly express that in HCT116, the effect is the opposite. Therefore, Fig3A would be better expressed as to what miRNAs are under- or overexpressed, and not as "identified" as a pool of DEG.
Additionally, I appreciate the authors giving the following explanation in their answer:
The possible reason is that the INO80 complex, as a chromatin remodeling complex, is widely involved in the transcriptional regulation of multiple genes by altering chromatin structure. Upon knocking down the INO80 complex in different types of cancer cells, the proteins (or complexes) or transcription cofactors recruited by INO80 in the promoter region of the miR-371-373 gene cluster may have different effects on the expression level of miR-372-373; Furthermore, according to the currently reported literature, the expression level of miR-372 varies depending on the type of cancer. For example, compared to normal tissues, miR-372-373 is significantly reduced in tissues such as non-small cell lung cancer, pancreatic cancer, cervical cancer, endometrial cancer, and ovarian cancer. On the contrary, the expression of miR-372 is higher in lung squamous cell carcinoma, colorectal cancer, esophageal squamous cell carcinoma, and hepatocellular carcinoma than in normal tissues. It is worth mentioning that in certain specific cancer types, the expression level of miR-372 contradicts the results reported by different research groups, such as breast cancer and colon cancer. Under our experimental conditions, we found that miR-372 expression was significantly reduced in colon cancer tissue compared to normal tissue. The expression status of miRNAs in different cancer cells may be closely related to the target genes they interact with, highlighting the complexity of the molecular mechanisms of miRNAs in tumor development in different cancer.
However, I think that this explanation should not be only for the reviewer, but for the general scientific audience reading the publication. Therefore, I suggest that the authors integrate also this piece within the Discussion section.
Author Response
The authors are thankful to the reviewer for understanding the purpose of this manuscript, making comments, and giving valuable suggestions to improve the quality of the manuscript.

Round 4
Reviewer 3 Report
ok